# DISTRIBUTION SHIFT AWARE NEURAL FEATURE TRANSFORMATION

## ABSTRACT

Feature transformation, as a core task of Data-centric AI (DCAI), aims to improve the original feature set to enhance AI capabilities. In dynamic real-world environments, where there exists a distribution shift, feature knowledge may not be transferable between data. This matter prompts a distribution shift feature transformation (DSFT) problem. Prior research works for feature transformation either depend on domain expertise, rely on a linear assumption, prove inefficient for large feature spaces, or demonstrate vulnerability to imperfect data. Furthermore, existing techniques for addressing the distribution shift cannot be directly applied to discrete search problems. DSFT presents two primary challenges: 1) How can we reformulate and solve feature transformation as a learning problem? and 2) What mechanisms can integrate shift awareness into such a learning paradigm? To tackle these challenges, we leverage a unique Shift-aware Representation-Generation Perspective. To formulate a learning scheme, we construct a representation-generation framework: 1) representation step: encoding transformed feature sets into embedding vectors; 2) generation step: pinpointing the best embedding and decoding as a transformed feature set. To mitigate the issue of distribution shift, we propose three mechanisms: 1) shift-resistant representation, where embedding dimension decorrelation and sample reweighing are integrated to extract the true representation that contains invariant information under distribution shift; 2) flatness-aware generation, where several suboptimal embeddings along the optimization trajectory are averaged to obtain a robust optimal embedding, proving effective for diverse distribution; and 3) shift-aligned pre and post-processing, where normalizing and denormalizing align and recover distribution gaps between training and testing data. Ultimately, extensive experiments are conducted to indicate the effectiveness, robustness, and trackability of our proposed framework. Our code is available at https://tinyurl.com/OODFT.

## 1 INTRODUCTION

In the modern era of machine learning, models are large and effective yet GPUs are expensive. Instead, Data-centric AI (DCAI) has emerged as an alternative solution by using AI to augment data power for better AI, even with simple models (Zha et al., 2023). As a core task of DCAI, feature transformation aims to transform

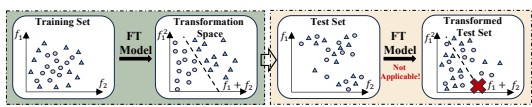

Figure 1: An illustration of DSFT, where the distribution of the training and test sets differ.

an original feature set into a more powerful one (Kusiak, 2001; Ying et al., 2023). In most cases, we assume that training and testing data are from the same domains and are I.I.D. without distribution shifts. In open environments, feature distributions, even from the same domain, can shift from training to testing data. Feature knowledge learned from data before shifts may not apply to data after shifts (see Figure 1). This case can be generalized as a distribution shift feature transformation (DSFT) problem. Addressing DSFT can enhance the generalization, robustness, applicability, and availability of feature transformation.

Relevant works can only partially solve DSFT. Firstly, DSFT is related to feature transformation. For instance, humans can manually transform a feature set with domain knowledge and empirical experiences. Machine-assisted methods include principal components analysis (PCA) (Maćkiewicz

& Ratajczak, 1993), exhaustive-expansion-reduction approaches (Kanter & Veeramachaneni, 2015; Khurana et al., 2016), iterative-feedback-improvement approaches (Khurana et al., 2018; Tran et al., 2016). However, human-based methods are usually time-consuming and incomplete. Among machine-assisted methods, PCA is only based on a straight linear feature correlation assumption, ignoring non-linear feature interaction, and PCA only reduces dimensionality instead of increasing dimensionality; others are mostly based on a discrete search formulation with critical limitations: 1) large search space and time costly when finding optimal, 2) lacks computational generalization and robustness against imperfect data, such as distribution shifts. Second, DSFT is related to anti-shift learning, generalization, and adaptive learning, for instance, distribution alignment (Fan et al., 2024; Kim et al., 2021; Hu et al., 2023), parameter isolation (Zhang et al., 2023), adaptive experience replay (Li et al., 2024). A clear research gap is that it is difficult to integrate shift awareness in learning-based formulation into discrete search-based formulations.

There are two major challenges to bridge the gap between feature transformation and shift awareness: 1) **From search-based formulation to learning-based formulation.** Many existing studies view feature transformation as a task of searching for the best feature set among large discrete feature combination possibilities. The first challenge seeks to answer: what paradigm can solve feature transformation as a learning problem: within a continuous optimization space, measured by a discriminative or generative model, evaluated by a tangible objective, guided by directional gradient? 2) **From shift-sensitive to shift-robust.** Once the feature transformation learning paradigm is proposed, the second challenge intends to answer: what mechanisms can integrate shift awareness into such a learning paradigm?

**Our Insights: A Shift-aware Representation-Generation Perspective.** We formulate the DSFT problem into a deep learning task, instead of a discrete search task. The study in (Wang et al., 2023) found that by collecting transformed feature sets and task performances as training data, we can embed transformed feature sets as embedding vectors. This continuous embedding space of transformed feature sets, if accurate, should include and describe the embedding point of the unobserved best transformed feature set at a certain maximum. Guided by this finding, we propose to regard searching the best feature transformation as a gradient optimization problem in the embedding space, in order to identify the optimal embedding. To address the learning framework challenge, we develop a representation-generation framework: 1) the representation step encodes transformed feature sets into embedding vectors; 2) the generation step identifies the best embedding via gradient ascent and decodes the best embedding to the best transformed feature set. Most importantly, this framework, although widely used in computer vision, opens up three essential opportunities for integrating shift awareness into representation, generation, and pre/post-processing for an old-school feature engineering problem.

To address the distribution shift, we develop the following three mechanisms in response to shift-resistant representation, flatness-aware generation, and shift-aligned pre and post-processing. (i) Connecting embedding dimension decorrelation with sample reweighing can incorporate shift resistance into embedding. (ii) Leveraging flatness in gradient-based optimal embedding search can ensure that the performance of a transformed feature set embedding does not decrease significantly in a neighborhood around the maximum, thus mitigating distribution shift. (iii) Performing normalization in pre-processing and denormalization in post-processing can align distribution gaps between training and testing data.

**Summary of Proposed Solution.** Inspired by these insights, *we propose a principled representation-generation-based deep feature transformation learning framework*, where the representation step trains an encoder to encode transformed feature sets into embedding vectors, by optimizing an evaluator (whether the embedding of a transformed feature set can be used to predict its corresponding performance) and a decoder (whether a transformed feature set can be reconstructed given its corresponding embedding), and the generation step exploits gradient ascent to identify the best transformed feature set embedding, and use the well-trained decoder to decode the best embedding and generate the best transformed feature set. This framework's significance lies not in its novelty, but in its ability to offer flexibility, thereby facilitating awareness shifts across three channels. Specifically, to achieve shift-resistant embedding, we reformulate a feature set as a feature-feature similarity graph and develop a new graph neural network that jointly considers partial covariance matrix norm minimization, sample reweighing in evaluator loss residuals, and decoder optimization, and is optimized by a bilevel training strategy. To achieve flatness-aware feature transformation generation, we connect oscillating search and suboptimal embedding averaging

to achieve a flat region in optimal embedding identification for the robust transformed feature set decoding. Finally, we leverage normalization and denormalization in pre and post-processing to further alleviate shifts.

**Our contributions** are: 1) we formulate DSFT as a continuous optimization problem and employ a representation-generation scheme to identify the optimal transformed feature space; 2) we propose shift-resistant feature set representation, flatness-aware generation, and integration of normalization to address distribution shift; 3) we conduct extensive experiments to demonstrate the efficiency, resilience, and traceability of our framework.

## 2 PROBLEM STATEMENT

**The DSFT Problem.** Given a feature set, a training dataset, and a test dataset, and both training and test data share the same feature set, considering the existence of distribution shifts between training and test data samples, we aim to learn an anti-shift feature transformation model to transform the original shifted training and test datasets into a transformed training dataset and a transformed test dataset, in order to: 1) improves the performance of an ML task (e.g., regression, classification); 2) is anti-shift. Our perspective is that the knowledge about what feature structure advances data power can be learned from historical exploration experiences of feature transformations and performances. We convert the DSFT problem as a shift-aware generative feature transformation learning task that involves several concepts:

**1) Feature Cross.** We apply operations to features, generating feature cross (e.g., $f_1 + f_1/f_3 - f_2$).

**2) Feature Transformation Operation Sequence.** We combine multiple feature crosses to construct a new transformed feature set (e.g., $f_1 + f_1/f_3 - f_2, \sqrt{f_1}$).

**3) The postfix expression of feature transformation operation sequence.** We use a postfix expression to represent the feature transformation operation sequence. For instance, the postfix expression of $f_1 + f_1/f_3 - f_2, \sqrt{f_1}$ is: $SOS f_1 f_1 f_3 / + f_2 - SEP f_2 \sqrt{.} EOS$. The postfix expression reduces token redundancy, prevents illegal transformation operations and semantic ambiguity, and minimizes search space. After converting a transformed feature set into a postfix token expression of a feature transformation operation sequence, DSFT task can be solved as a sequential generation task that generates a feature transformation operation sequence.

## 3 RL AGENTS FOR DATA COLLECTION

Before modeling, we collect various transformed feature sets and their corresponding performances (e.g., accuracy) on a specific downstream task (e.g., regression, classification) as training data.

**Why automated and diverse training data collection.** To learn a feature set2vec encoder, we need to collect various feature sets and their accuracy on a downstream task as a supervised knowledge base. Intuitively, we can manually and randomly try different possible combinations of feature crosses (e.g., $f_1 + f_2, f_1/f_2$. Here, $f_1$ is a head feature selected by a head agent, and $f_2$ is a tail feature selected by a tail agent. $f_1$ and $f_2$ can represent any feature.) to form a new feature set, and then test the performance of the new feature set on a controlled regression or classification task. However, the three aspects of training data are essential: 1) *large volume*: sufficient training data; 2) *high quality*: high-accuracy feature set cases as successful experiences; 3) *high diversity*: training data should not ignore random, exploratory, and failure cases.

**Leveraging reinforcement learning to explore high-quality and diverse training data.** Inspired by the exploitation, exploration, and self-learning abilities of reinforcement learning, our idea is to view an RL agent as a training data collector in order to achieve volume (self-learning enabled automation), diversity (exploration), and quality (exploitation). Specifically, we design RL agents to automatically decide

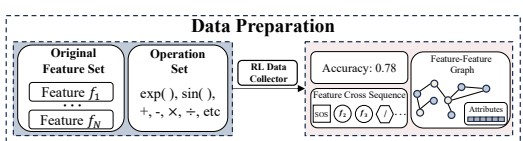

Figure 2: Data preparation pipeline. RL agents are employed to collect exemplary and diversified training data.

how to perform feature cross and generate new feature sets. The reinforcement exploration experiences and corresponding accuracy will be collected and stored as training data as shown in Figure 2. More details about the RL data collector are described in Appendix A.

# 4 THE REPRESENTATION-GENERATION FRAMEWORK

## 4.1 FRAMEWORK OVERVIEW

Figure 3 shows our Shift Awareness Feature Transformation framework (SAFT) includes three components: 1) shift-resistant feature set representation; 2) flatness-aware feature transformation generation; 3) integrated prenormalization and post-denormalization. To achieve shift-resistant representation, we develop two insights: 1) feature sets as dynamic feature-feature interaction graphs and 2) a combination of sample reweighing and embedding dimension orthogonality for invariant representation. To achieve flatness-aware feature transformation generation, we integrate flatness-aware gradient ascent, which aims to identify a flatter optimal embedding for shift robustness. Finally, we integrate normalization in prepro-

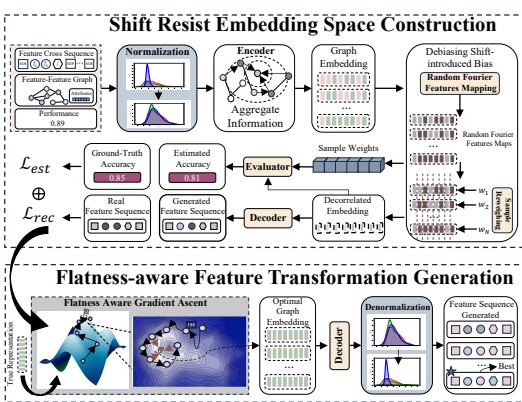

Figure 3: An overview of our framework.

cessing and denormalization in postprocessing to further alleviate the impacts of shifts. The Algorithm 1 in Appendix B.1 provides the detailed pseudo-code.

## 4.2 SHIFT-RESISTANT FEATURE GRAPH EMBEDDING

To find the best transformed feature sets, our key insight is to effectively represent feature sets and corresponding performances in an embedding space. Using training data composed of feature sets and their predictive performances, we train an encoder and a decoder. These components map transformed feature sets to vectors and vice versa, creating a continuous embedding space. We believe the best transformed feature set, termed feature knowledge, is described in this continuous embedding space and, thus, can be efficiently identified by gradient optimization. There is no need to explore exponentially growing possibilities of feature combinations. Although training-testing shifts may distort the embedding space, the space is learnable. Anti-shift mechanisms can rectify and adjust the distortion and ensure reliable and transferable feature knowledge in the testing set.

*A. Encoder for Learning Dynamic Feature-Feature Interaction Graph Embedding.* Representing a discrete feature set is traditionally challenging. Conventional methods either aggregate the first-order or second-order statistics of each feature or regard a set of features as a token sequence for encoding. However, a feature set is not a decorrelated independent set. Features within a set strongly associate to demonstrate distinctive patterns and enhance ML performances. When ignoring feature-feature interaction, feature set representations are likely to be biased and inaccurate, which introduces errors in the optimal feature set identification.

Our insight is to view a feature set as a feature-feature interaction attributed graph, where a feature is a node, the element values of a feature are node attributes, and feature-feature similarity (i.e., cosine similarity) is the weight of an edge. After converting feature sets into graphs, existing graph embedding methods, for example, Graph Convolutional Networks (GCNs), could have been applied to map a feature set to an embedding vector. However, we observe that the sizes (feature numbers) and topology (adjacency matrices) of feature-feature similarity graphs vary a lot. Classic GCNs are unable to address dynamic graph structure (sizes and topology) because GCNs require fixed graph sizes and topology. Inspired by (Hamilton et al., 2017), we tailor the idea of node sampling and neighbor information propagation from increasingly distant reaches for dynamic feature-feature graph embedding. Specifically, given a feature-feature similarity attributed graph at each step, each node consolidates the previous representations of itself and its neighbor representations. A fully connected layer with a nonlinear activation function is later utilized to update the current representation. We normalize the embedding for each step and iterate to obtain the final representation.

*B: Estimator for Estimating Feature Set Performances.* Our goal is to learn an embedding space to represent feature sets so that we can find the optimal feature set embedding with higher performance, which is used to reconstruct the optimal feature set. To strengthen the performance expressiveness of feature set embedding, we incorporate a feature set performance estimator as a downstream task of the feature-feature graph encoder. Specifically, we employ a feedforward network to take a feature-

feature graph embedding as input and predict the performance of a feature set, by minimizing the mean square error between estimated feature set performances and ground-truth feature set performances. Formally, the loss of evaluator is $\mathcal{L}_{est} = \frac{1}{N}\sum_{i=1}^{N}(p_i - \omega_\theta(E_i))^2$, where $N$ is total number of feature sets in training data, $i$ index each feature set, $p_i$ is the ground-truth feature set performance, $\omega_\theta$ is the feature set performance estimator given a feature set's graph embedding, $E_i$ is the sample embedding.

*C: Decoder.* The primary objective of the decoder is to output a feature transformation operation sequence (deemed as a token sequence) given a feature-feature graph embedding. The decoder provides two main functionalities: 1) enforcing the faithfulness of embedding: the embedding of a transformed feature set should be able to fully reconstruct its corresponding feature transformation operation sequences; 2) generates the optimal transformed feature set: a well-trained decoder can be used to generate the feature transformation operation sequence of the optimal embedding. Our decoder is structured by a single-layer LSTM and a softmax layer, where the LSTM is to recursively learn the hidden state over the current-previous dependent feature transformation operation sequence, and the softmax layer is to estimate the token probability when generating each token. Specifically, given $\psi$ as a single-layer LSTM, the distribution of the single $i$-th token is $P_\psi(\gamma_i|E, \Upsilon_{<i}) = \frac{\exp(s_j)}{\sum_M \exp(s)}$. where, $\Upsilon$ is a feature transformation operation token sequence with $M$ token length, $\gamma_i$ is the $i$-th token in $\Upsilon$, and $s_j$ is the $j-\text{th}$ output of the softmax layer. Finally, we aim to minimize the negative log-likelihood of decoding a feature transformation operation token sequence as the reconstruction loss: $\mathcal{L}_{rec} = -\sum_N \sum_{i=1}^{M} log P_\psi(\gamma_i|E, \Upsilon_{<i})$.

*D: Shift-resistant Joint Optimization of Encoder, Estimator, and Decoder.* We will develop a shift-resistant optimization strategy to learn the encoder, estimator, and decoder.

**The Joint Objective Function.** The separate training of the encoder, estimator, and decoder can lead to local optima. Instead, we optimize the joint loss of the encoder, evaluator, and decoder. Our objective is: $\mathcal{L}_{est} + \gamma \mathcal{L}_{rec}$, where $\gamma$ are a hyper-parameter. In this way, we enforce the encoder-evaluator-decoder structure to learn a faithful embedding to accurately predict the performance of a transformed feature set and decode the corresponding feature transformation operation sequence.

**Debiasing Shift-induced Biases in Training for Shift-aware Learning.** In open environments, there exist distribution shifts between training and testing data. The shifts introduce bias into the embedding space learned from training data, which can not be generalized to shifted test data. The learned biased embedding thus includes not just true representations, but also false representations. The true representations are relevant and genuine descriptors of the performance of a transformed feature set, and remain invariant under distribution shifts. The false representations are irrelevant descriptors that drift from training to testing. The subtle and fake correlations between false representations and true representations mislead the joint loss optimization process to relate false representations to the labels of feature set performances, thus, degrading generalization abilities under distribution shifts.

To alleviate the negative influence of false representations, an intuitive idea is to decorrelate true and false representations during learning. However, true and false representations are unknown and not pre-identified. We are unable to directly decorrelate true and false representations. A conservative and aggressive solution is to decorrelate all representations by encouraging the orthogonality of all embedding dimensions. This objective can be reformulated into the minimization of the squared Frobenius norm of a partial covariance matrix of the embedding dimensions. This Frobenius norm can be added as a loss term into the joint loss minimization. However, 1) it relies on the aggressive and strong assumption on the structure of embedding dimensions, thus, suffers from loss of information; 2) even though all embedding dimensions are decorrelated, the false representations are still in the embedding. In other words, feature set embeddings are still not pure and suffer from the inclusion of false and biased dimensions, resulting in low generalization over changing distributions.

Our goal is to debias the embedding bias caused by distribution shift so the embeddings include the true invariant correlations between true representations and feature set performances. We connect and unify two interesting insights: 1) A recent study (Shen et al., 2020) shows that there exists a set of sample weights that can reshape the sample-variable matrix and make a weighted covariate distribution matrix near orthogonal. 2) Another study (Athey et al., 2018) shows that reweighing the losses of training samples of transformed feature sets can achieve a debiasing effect in selecting variables to rebuild representations for better generalization. Interestingly, this insight closely

connects to the boosting strategy that reweighs training samples (i.e., increase misclassified sample weights and lower correctly classified sample weights) to enforce the next weak classifier to avoid making mistakes in previous misclassified samples. In summary, training sample reweighing can be seen as model regularization. Inspired by the two findings and (Zhang et al., 2021; Li et al., 2022), we develop a unified bilevel optimization approach: 1) *the inner-loop level*: we see the sample weights as learnable parameters that are learned by minimizing the squared Frobenius norm of a partial covariance matrix; 2) *the outer-loop level*: we use the weights to reweigh the sample losses in estimating the performance of each feature-feature graph embedding. In this way, we leverage sample reweighing and bi-level training as a bridge to achieve both decorrelating all embedding dimensions and reducing false dimensions caused by the bias of distribution shift.

**The Shift-resistant Bilevel Training.** The Inner-loop Training. For the inner-loop, our objective is to learn the best sample weights to minimize the squared Frobenius norm as follows.

$$\mathbf{R}^* = \arg\min_{\mathbf{R}} \sum_{i<j} \|\hat{\mathbf{C}}^{\mathbf{R}}_{E_{*i},E_{*j}}\|^2_F, \tag{1}$$

where $\hat{\mathbf{C}}^{\mathbf{R}}_{E_{*i},E_{*j}} = \frac{1}{N-1}\sum_{n=1}^{N}[(r_n f(E_{ni}) - \frac{1}{N}\sum_{m=1}^{N} r_m f(E_{mi}))^\top \cdot (r_n g(E_{nj}) - \frac{1}{N}\sum_{m=1}^{N} r_m g(E_{mj}))]$. Here, $\hat{\mathbf{C}}^{\mathbf{R}}_{E_{*i},E_{*j}}$ is the partial cross-covariance matrix, $\mathbf{R} = \{r_n\}_{n=1}^{N}$ is the graph weight vector, $r_i$ is the weight of $i$-th feature subset graph and we constraint $\sum_{n=1}^{N} r_n = N$. $f(\cdot)$ and $g(\cdot)$ are the random Fourier features functions, $E_{*i}$ and $E_{*j}$ are the different dimensions of the same training sample.

The Outer-loop Training. We use the updated weights of training samples $\mathbf{R}^* = \{r_n^*\}_{n=1}^{N}$ to reweigh the estimation loss of each transformed feature set, given by $\mathcal{L}_{est} = \sum_{i=1}^{N} r_i^*(p_i - \omega_\theta(E_i))^2$. Then, we combine the new estimator loss and decoder loss $\mathcal{L}_{rec}$ to obtain the weighted joint loss: $argmin \mathcal{L} = \mathcal{L}_{est} + \gamma\mathcal{L}_{rec}$. The Algorithm 2 in Appendix B.2 provides the detailed pseudo-code.

### 4.3 FLATNESS-AWARE TRANSFORMATION GENERATION

After learning the embedding space of transformed feature sets, the embedding space should accurately describe and represent all transformed feature sets, including the unobserved best transformed feature set. Hence we utilize gradient ascent to identify the best embedding vector with the highest performance to decode.

**Step 1: Gradient-ascent Optimization to Identify The Optimal Feature Set Embedding** Another benefit of learning an embedding performance evaluator in the representation step is to make differentiable gradient optimization possible. Particularly, we extract the gradient from the evaluator toward the direction of maximizing feature set performance. Formally, the gradient-ascent is defined by : $\hat{E} = E + \eta\frac{\partial\omega_\theta}{\partial E}$, where $\omega_\theta$ is the evaluator, $\hat{\mathbf{E}}$ is the optimal embedding, $\eta$ is the step size. In the experiments, we select top-T transformed feature set embeddings in training data as the initialization seeds of gradient ascent. Therefore, we can identify T-improved embedding vectors as the optimal embedding set $\hat{\mathcal{E}} = \{\hat{E}^t\}_{t=1}^{T}$.

**Step 2: Incorporating Flatness-aware into Gradient Ascent to Mitigate Shifts.** In gradient optimization, the flatness of the loss landscape has been proven to exhibit a close connection with distribution shift resistance both theoretically and empirically. Under an open world, when there exists a distribution shift between training and testing data, the optimal transformed feature set embedding on test samples does not coincide with the optimal transformed feature set embedding found on training samples. Flatness ensures that the loss does not increase significantly in a neighborhood around the found minimum. Therefore, flatness leads to distribution shift resistance because the loss on test examples does not increase significantly. Inspired by (Izmailov et al., 2018; Garipov et al., 2018), we leverage the advantage of loss flatness and develop a flatness-aware gradient ascent approach.

Specifically, unlike classic gradient ascent, we propose to enforce a searching process to oscillate around the optimal embedding to collect more suboptimal embedding vectors. This suboptimal embedding set can pinpoint a flat region where the real optimal point for the test set is located. We then aggregate all the suboptimal points as the final averaged embedding that represents the center of the flat region in the loss landscape. However, averaging all the suboptimal points of each gradient ascent iteration will introduce huge computational costs and include the non-optimal points, which

are far away from the optimal neighborhood. We therefore develop a cyclic scheme, in which a cycle includes multiple gradient ascent iterations, and we only average the suboptimal points at the end of a cycle. To avoid missing the most optimal point when approaching the maximum, we utilize linearly decreasing learning rates within each cycle. In particular, Algorithm 3 shows we linearly decrease the learning rate $\eta(i)$ from $\eta_1$ to $\eta_2$ over iterations during a cycle. In each iteration of a cycle, we initialize the learning rate and update the embedding by gradient ascent. At the end of each cycle, we average the embedding (Line 7).

**Step 3: Decoding Embeddings to Reconstruct Optimal Feature Reconstruction Operations** After obtaining the candidate embedding set $\hat{\mathcal{E}}$, we use the well-trained decoder $\psi$ to generate the feature transformation operation sequences, i.e., $\hat{\mathcal{E}} \xrightarrow{\psi} \{\hat{\Upsilon}_i\}_{i=1}^T$. Specifically, the decoder iteratively generates the next token of a feature transformation operation sequence, such as a feature token, an operator token, or a segmentation token 'SEP' in an autoregressive manner until it produces an end of sequence token 'EOS'. Finally, we divide the generated sequence into multiple segments using the 'SEP' token and transform each segment into a feature, resulting in the optimal transformed feature set with the best estimated performance.

### 4.4 Integrating Normalization-Denormalization into Pre- and Post-Processing

In addition to embedding dimension orthogonality and flatness-aware optimization, we propose a data-centric perspective to alleviate distribution shift by first aligning and then recovering the distribution gaps between training and test data in pre and post-processing. We propose to incorporate a normalization and denormalization mechanism in pre and post-processing. Specifically, we will first normalize (e.g., z-score) original training and testing data before feature transformation, to obtain normalized training and testing data with zero mean and one standard deviation. We then train our method with the normalized training data to generate the optimal feature transformation operation sequence, under a normalized distribution. Later, we apply the optimal feature transformation operation sequence learned from the normalized distribution to transform the normalized testing data. So we obtain the transformed test data under the normalized distribution. Finally, we denormalize (e.g., reverse z-score normalization) the transformed testing data to obtain the transformed testing data in the original distribution. The underlying idea is to leverage normalization-denormalization to conduct feature transformation learning in a normalized distribution and avoid shift-caused bias.

## 5 Experimental Results

### 5.1 Experimental Setup

**Datasets.** We perform experiments on 16 publicly available datasets from UCI (Dua & Graff, 2017) and OpenML (Vanschoren et al., 2013). 9 datasets are intended for regression, and the remainder parts are for classification. To evaluate the robustness of SAFT against distribution shifts, we iteratively generate training and testing sets, employing the Kolmogorov-Smirnov test to identify distribution shifts. Once a shift is detected, we finalize the training and testing sets with allocations of 80% and 20%, respectively. More details about the data selection are included in Appendix C.1.

**Baselines.** We compare our framework with 8 wildly-used feature transformation methods. We described the details about all baselines in Appendix C.2.

**Evaluation Metrics And Hyperparameters Setting.** To control the variance of the downstream model impact on evaluation, we apply Random Forests (RF) for both classification and regression tasks. We described more details in Appendix C.3 and C.4.

**Environmental Settings.** All experiments are conducted on the Ubuntu 22.04.3 LTS operating system, Intel(R) Core(TM) i9-13900KF CPU@ 3GHz, and 1 way RTX 4090 and 32GB of RAM, with the framework of Python 3.11.4 and PyTorch 2.0.1.

### 5.2 Performance Results

In this experiment, we compare the feature transformation performance of SAFT with other baseline models. Table 1 shows the overall comparison results in terms of F1 score and 1-RAE. We observe that SAFT consistently outperforms other baseline models across all datasets. The underlying driver for this observation is that SAFT can compress the feature learning knowledge into a robust embedding space through shift-resistant embedding learning, and smoothly search for the optimal feature

Table 1: The overall performance results across various real-world datasets. The best and second-best outcomes are indicated by bold and underlined fonts, respectively. We measure the performance on classification (C) and regression (R) tasks using F1-score and (1-RAE) metrics, respectively. A higher value indicates a superior quality of the feature transformation space.

| Dataset | C/R | Samples | Features | RDG | ERG | LDA | AFAT | NFS | TTG | GRFG | MOAT | SAFT |
|---|---|---|---|---|---|---|---|---|---|---|---|---|
| Housing Boston | R | 506 | 13 | 0.375 | 0.366 | 0.146 | 0.387 | 0.395 | 0.383 | 0.361 | 0.395 | **0.405** |
| Airfoil | R | 1503 | 5 | 0.733 | 0.695 | 0.522 | 0.742 | 0.742 | 0.738 | 0.614 | 0.724 | **0.743** |
| openml_586 | R | 1000 | 25 | 0.542 | 0.536 | 0.104 | 0.540 | 0.543 | 0.543 | 0.334 | 0.616 | **0.649** |
| openml_589 | R | 1000 | 50 | 0.509 | 0.472 | 0.099 | 0.467 | 0.470 | 0.469 | 0.436 | 0.496 | **0.582** |
| openml_607 | R | 1000 | 50 | 0.306 | 0.310 | 0.054 | 0.344 | 0.358 | 0.354 | 0.371 | 0.424 | **0.516** |
| openml_616 | R | 500 | 50 | 0.197 | 0.311 | 0.014 | 0.342 | 0.344 | 0.343 | 0.459 | 0.369 | **0.534** |
| openml_618 | R | 1000 | 50 | 0.421 | 0.391 | 0.058 | 0.436 | 0.430 | 0.431 | 0.257 | 0.427 | **0.468** |
| openml_620 | R | 1000 | 25 | 0.510 | 0.464 | 0.025 | 0.475 | 0.464 | 0.462 | 0.495 | **0.566** | 0.545 |
| openml_637 | R | 500 | 50 | 0.265 | 0.340 | 0.015 | 0.365 | 0.344 | 0.339 | 0.380 | 0.381 | **0.424** |
| Higgs Boston | C | 50000 | 28 | 0.693 | 0.694 | 0.508 | 0.692 | 0.691 | 0.696 | 0.698 | 0.702 | **0.704** |
| SpectF | C | 267 | 44 | 0.674 | 0.792 | 0.651 | 0.643 | 0.716 | 0.672 | 0.728 | 0.766 | **0.799** |
| UCI Credit | C | 30000 | 25 | 0.809 | 0.808 | 0.743 | 0.805 | 0.805 | 0.803 | 0.797 | 0.808 | **0.816** |
| Wine Quality Red | C | 999 | 12 | 0.658 | 0.491 | 0.588 | 0.662 | 0.650 | 0.675 | 0.668 | 0.681 | **0.700** |
| Wine Quality White | C | 4900 | 12 | 0.730 | 0.726 | 0.602 | 0.715 | 0.724 | 0.718 | 0.680 | 0.730 | **0.734** |
| PimaIndian | C | 768 | 8 | 0.679 | 0.761 | 0.685 | 0.636 | 0.691 | 0.734 | 0.689 | 0.763 | **0.780** |
| Geman Credit | C | 1000 | 24 | 0.697 | 0.662 | 0.693 | 0.656 | 0.698 | 0.683 | 0.647 | 0.707 | **0.743** |

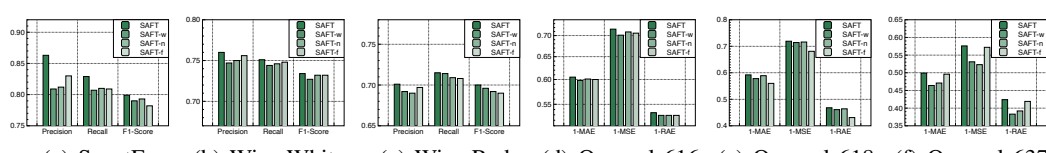

(a) SpectF    (b) Wine White    (c) Wine Red    (d) Openml_616    (e) Openml_618    (f) Openml_637

Figure 4: Results on the impact of normalization, Flatness Aware Gradient Ascent, and reweighting.

space via flatness-aware weight averaging. Moreover, we notice that the second-best model is variant among different cases. A potential reason for this observation is that baseline approaches overlook the impact of distribution shifts, resulting in an unstable identification of the optimal feature space. This experiment underscores that SAFT effectively captures invariant knowledge in feature learning against distribution shifts and produces a robust transformed feature space.

## 5.3 ABLATION STUDY

To evaluate the necessity of various components in SAFT, we develop three model variants: 1) SAFT-f, which excludes the Flatness Aware Gradient Ascent without performing weight averaging; 2) SAFT-n, which omits the normalization process; 3) SAFT-w, which removes the optimization of graph weights. We select three classification and three regression datasets for conducting experiments. Figures 4 show the comparison results. First, we observe that across all situations, the performance of the three model variants shows a decline when compared to SAFT. A potential reason is that disregarding strategies aware of distribution shifts fails to consider the variation between training and testing sets, resulting in suboptimal feature transformation performance. Moreover, SAFT can perform better compared to SAFT-w. This observation indicates that the learned graph weights can eliminate spurious correlations among features, thereby enhancing the transformed feature space. Additionally, we find that SAFT outperforms SAFT-f. The underlying driver is that Flatness Aware Gradient Ascent prevents the gradient search process from converging to local optimal points, thereby smoothing and making the search process more robust. Furthermore, SAFT-n is inferior to SAFT. This observation reflects that aligning statistical properties via normalization is an effective strategy for tackling the OOD issue in feature space. Thus, these experiments demonstrate that each technical component of SAFT is indispensable for mitigating the impact of distribution shift on feature transformation.

## 5.4 ROBUSTNESS ANALYSIS

**Dataset Split Robustness.** To check the robustness of SAFT for different data splits, we use five distinct splits on the Wine Quality White dataset for evaluation. Each split defines a unique 20% proportion of the dataset as the testing set, moving sequentially through the dataset from beginning to end, with the remainder serving as the training set.

Figure 5 presents the comparison results in terms of F1-score, Precision, and Recall.

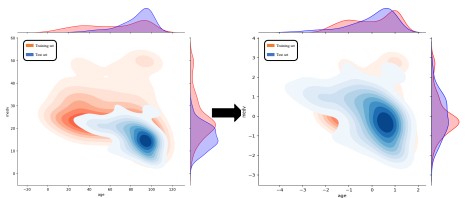 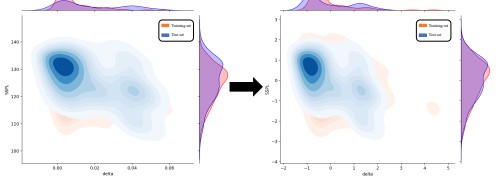

Figure 7: Comparison of the changes in joint and marginal distributions of age and medv (target) in the Housing Boston dataset for both training and test sets. Left: Before normalization. Right: After normalization.

Figure 8: Comparison of the changes in joint and marginal distributions of delta and SSPL (target) in the Airfoil dataset for both training and test sets. Left: Before normalization. Right: After normalization.

We notice that despite potential distribution shifts introduced by different dataset split approaches, SAFT still demonstrates robust performance in addressing these shifts. The main reason for this observation is that SAFT can preserve the invariant and robust feature learning knowledge within the embedding space for robust and smooth search. Such a learning mechanism can alleviate the impact of distribution shifts in different data splits. Hence, this experiment shows that SAFT exhibits robust performance in addressing distribution shift, regardless of the dataset split methods.

Table 2: Robustness check on ML models for the Wine Quality White dataset.

|      | RF    | SVM   | KNN   | DT    | LASSO | Ridge |
|------|-------|-------|-------|-------|-------|-------|
| RDG  | 0.658 | 0.632 | **0.629** | 0.630 | 0.634 | 0.606 |
| ERG  | 0.491 | 0.354 | 0.358 | 0.413 | 0.470 | 0.522 |
| LDA  | 0.588 | 0.439 | 0.512 | 0.579 | 0.390 | 0.410 |
| AFAT | 0.662 | 0.582 | 0.443 | 0.663 | 0.596 | 0.587 |
| NFS  | 0.650 | 0.634 | 0.542 | 0.639 | 0.634 | 0.601 |
| TTG  | 0.675 | 0.644 | 0.525 | 0.602 | 0.630 | 0.620 |
| GRFG | 0.668 | 0.313 | 0.502 | 0.616 | 0.551 | 0.540 |
| SAFT | **0.700** | **0.655** | 0.556 | **0.677** | **0.659** | **0.658** |

**Downstream Performance Robustness.** To check the robustness of SAFT for distinct downstream ML models, we substitute the downstream model with Random Forest (RF), Support Vector Machine (SVM), K-Nearest Neighborhood (KNN), Ridge, LASSO, and Decision Tree (DT). Table 2 shows the comparison results on the Wine Quality Red dataset in terms of the F1-score. We find that SAFT outperforms baselines in most cases. A potential reason for this observation is that SAFT captures invariant and generalizable feature learning knowledge, leading to its superior generalization across various downstream ML models. Thus, this experiment demonstrates the robustness of SAFT across various ML models.

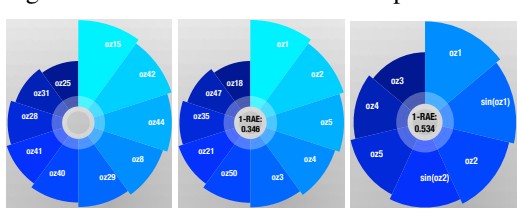

(a) Precision    (b) Recall    (c) F1 Score

Figure 5: Robustness of the dataset split methods.

(a) Training Set    (b) Test Set    (c) SAFT

Figure 6: Comparison of the most important features in the original feature space of training and test set, and the SAFT generated feature space.

### 5.5 Qualitative Analysis

**Shift Elimination.** We perform a case study to demonstrate the impact of normalization in alleviating the distribution shift between the training and test sets. Figures 7 and 8 indicate the changes in the distribution of the Housing Boston and Airfoil datasets, respectively. It is evident that, after normalization, the joint and marginal distributions of age and medv (target) in both the training and test sets exhibit significant alignment. While the joint distribution of delta and SSPL (target) remains relatively stable, the marginal distributions of the training and test sets show alignment. This case study shows that normalization is effective in mitigating the distribution differences between the training set and test set.

**Feature Importance.** We choose the top 10 most essential features from the original training set, the original test set, and the transformed feature space using SAFT for the openml_616 dataset to facilitate comparison. The importance of features is measured by the mutual information (MI) between the features and the target. Figure 6 displays the results. The labels accompanying each pie indicate the respective feature names, with larger areas signifying greater importance. We observe

that the critical features in the training set differ from those in the test set, attributable to the distribution shift. Consequently, optimal feature combinations in the training set may not directly apply to the test set, as their significance may vary. Furthermore, we note that in the new feature space generated by SAFT only 7 features are employed (with 2 new features generated by SAFT), but these features improve the performance of ML model by 29.24%. This phenomenon illustrates that SAFT can identify and generate effective features that are applicable to the test set. SAFT can capture the true representation of the feature set and generate a robust embedding space.

To make the experiment more convincing, we also analyzed the time complexity and space complexity in Appendix D. Meanwhile, we visualize the embedding space to verify the effectiveness of the intermediate encoding in Appenddix E.

## 6 RELATED WORKS

**Feature Transformation** aims to construct an enhanced feature space by transforming original features. Previous research can be categorized into three categories: 1) *Expansion-reduction based approaches* (Kanter & Veeramachaneni, 2015; Horn et al., 2020; Khurana et al., 2016), where the original feature space is expanded through explicit or greedy mathematical transformations and subsequently the feature space is narrowed down by selecting valuable features. 2) *Evolution-evaluation approaches* (Wang et al., 2022; Khurana et al., 2018; Tran et al., 2016), where evolutionary algorithms or Reinforcement learning models are employed to optimize the process of iteratively creating effective features and retaining significant ones. 3) *Auto ML-based approaches* (Chen et al., 2019; Zhu et al., 2022; Wang et al., 2023), where the most appropriate model architecture is automatically identified to formulate AFT as an Auto ML task. However, these methods face two challenges: 1) high-order feature transformation is difficult to produce; 2) transformation performance is unstable. To address these obstacles, we formulate AFT as a continuous optimization task (Wang et al., 2023). In particular, we employ an RL-based data collector to prepare the training data and utilize an encoder-decoder evaluator-based architecture to build the continuous space. Then, we search for the optimum solution within the continuous space and reconstruct the transformed feature space.

**Distribution Shift Problems**, arising from inconsistent distributions between the training set and test set, is prevalent in the real-world domains. Existing works to solve this problem mainly focus on time series analysis (Fan et al., 2023), computer vision (Yu et al., 2023), and natural language processing (Dou et al., 2022), etc. But distribution shift problem for feature transformation is relatively underexplored and can greatly degrade the effectiveness of the reconstructed feature. To tackle this challenge, specifically, we integrate three techniques designed to function at different levels within our framework: 1) shift-resistant representation: where the invariant true representation is learned under DSFT; 2) flatness-aware generation: where a more robust embedding is pinpointed within the searching process; and 3) shift-aligned pre and post-processing: where distributions are aligned for the data processing.

## 7 CONCLUSION

In this work, we combine representation, generation, and anti-shift learning to design a robust feature transformation framework for DSFT, namely Shift Awareness Feature Transformation (SAFT) framework. Our contributions can be summarized as follows: First, to avoid searching in a large discrete, we formulate DSFT as a gradient optimization problem and develop a representation-generation framework to identify optimal transformed features. Second, to eliminate the effect of distribution shift, we integrate three mechanisms designed to operate at varying levels within our framework: 1) shift-resistant representation: to learn the invariant true representation under DSFT; 2) flatness-aware generation: to identify a more robust embedding in the searching process; and 3) shift-aligned pre and post processing: to align distributions for the data processing. Ultimately, the experimental results empirically demonstrate that even without the prior knowledge of test distribution, SAFT can generate a robust feature space and identify optimal transformed feature sequences based on the information learned from the training set. This emphasizes the superior generalization ability of SAFT to enhance AI capabilities in diverse open environments across domains, such as economics and health care.

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

## A    MORE DETAILS ABOUT DATA COLLECTION BY RL AGENTS

In this section, we provide more details on the RL agents used for data collection.

**Leveraging reinforcement learning to explore high-quality and diverse training data.** The training data collector includes: 1) Multiple Agents: We design three agents to perform feature crossing: a head feature agent, an operation agent, and a tail feature agent. 2) Actions: In each reinforcement iteration, the three agents collaborate to select a head feature, an operator (e.g., +,-,*,/), and a tail feature to generate a new feature (e.g. $f_1+f_2$). The newly generated feature is later added to the feature set for the next feature generation. To balance diversity and quality, we employ the $\epsilon$-greedy DQN algorithm. For each step, each agent has a $\epsilon$ probability of selecting a feature/operation based on policy and has a (1-$\epsilon$) probability of selecting a feature/operation equally. The $\epsilon$ will increase over time so that agents can first explore more and then exploit more. 3) Environment: The environment is the feature space, representing an updated feature set. When three feature agents generate and add a new feature to the previous feature set, the state of the feature space (i.e., the environment) changes. The state represents the statistical characteristics (such as aggregated average mean, and variance) of the feature subspace. 4) Reward Function: We formulate the reward as the improvement of the performance (e.g., accuracy) of the explored feature set on a downstream task (e.g., regression) on the current iteration, compared with that of the explored feature set on the previous iteration. In this way, we incentivize agents to explore high-quality feature sets. 5) Training and Optimization: Our reinforcement data collector includes many feature crossing steps. Each step consists of two stages - control and training. In the control stage, each feature agent takes actions to change the size and contents of a new feature set and generate a reward. This reward is assigned to each participating agent. In the training stage, the agents train their policies via experience replay independently by minimizing the mean squared error (MSE) of the Bellman Equation. 6) Feature Set-Performance Annotation Pairs Preparation: We test each RL-explored feature subset with a downstream ML task (e.g., random forest classification) to collect feature set performance annotation. We select the top 5,000 feature set-performance annotation pairs based on predictive accuracy as training data.

**Training data triples.** After obtaining the transformed feature set-performance pairs through RL, we convert this data into triples to build our training dataset. 1) We convert the transformed feature set into a feature-feature graph as input to the encoder. The purpose is to capture the structural relationships between features, enabling the encoder to better represent complex feature interactions. 2) The transformed feature set is treated as a feature transformation operation sequence. The purpose of obtaining this sequence is to allow the decoder to recognize and process it. The decoder then iteratively predicts one token at a time until it reaches an end-of-sequence (EOS) token. 3) The evaluator assesses the performance associated with the sequence, which guides the gradient-based search to identify the optimal embedding.

**The determination of the length of the feature transformation operation sequence.** We introduce a hyperparameter to control the number of transformed features. When this number exceeds the predefined limit, feature selection is applied to retain only the most important features. At each step, RL generates a new crossed feature, which is added to the original feature set as a candidate for the next iteration. Thus, the length of the sequence is influenced by the order in which the transformed features are added to the set. For example, two transformed feature sets have the same number of features but different lengths of sequence (e.g., $SOSf_1SEPf_2 + f_1EOS$ vs $SOSf_1SEP\sqrt{f_2}EOS$).

## B    MORE DETAILS ABOUT SHIFT AWARENESS FEATURE TRANSFORMATION FRAMEWORK (SAFT)

### B.1    ALGORITHM TABLE FOR THE OVERALL FRAMEWORK

This part provides an algorithm table for the overall framework (see Algorithm 1).

### B.2    ALGORITHM TABLE FOR THE SHIFT-RESISTANT BILEVEL TRAINING

In this part, we provide an algorithm table for the shift-resistant bilevel training (see Algorithm 2).

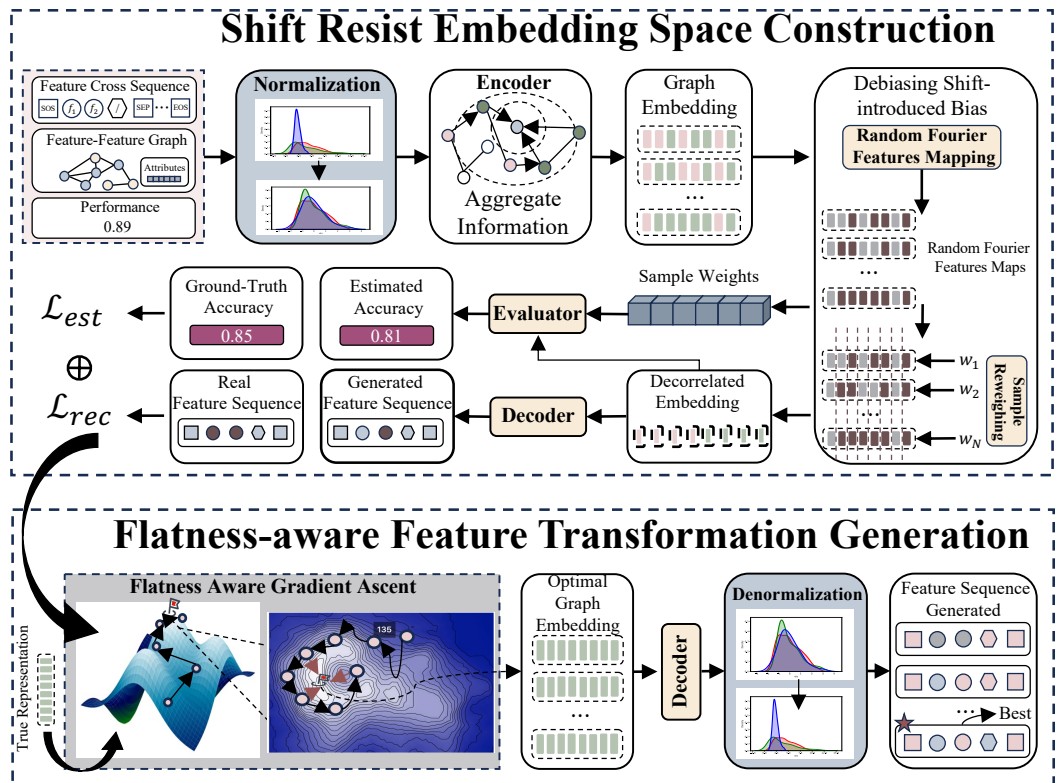

Figure 9: An overview of our proposed framework.

---

**Algorithm 1** Overall Framework
___
**Input:** Original dataset
**Output:** Optimal Transformed Feature set
 1: Initial the encoder, evaluator and decoder
 2: Leveraging RL agents to collect high quality data
 3: Construct feature-feature similarity graphs
 4: Normalize the collected data
 5: Algorithm 2 Shift-resistant Bilevel Training to train encoder, decoder, and evaluator
 6: Algorithm 3 to identify the optimal embedding
 7: Leverage well-trained decoder to generate the feature transformation operation sequences in an autoregressive manner based on the optimal embedding
 8: Denormalize the Optimal Transformed Feature set
___

### B.3 ALGORITHM TABLE FOR THE FLATNESS AWARE GRADIENT ASCENT

In this part, we provide an algorithm table for Flatness Aware Gradient Ascent(see Algorithm 3).

## C  MORE DETAILS FOR EXPERIMENTS SETUP

### C.1 MORE DETAILS FOR THE DATASET SELECTION

Starting from the first feature, we split the dataset into 80% for training and 20% for testing based on their observation index. Using the Kolmogorov-Smirnov test (Berger & Zhou, 2014) at a 95% confidence level, we identify distribution shifts. If detected, we finalize the training and testing sets with allocations of 80% and 20%. If no shift is detected, we move to the next feature. If no shift is detected across all features, we exclude the dataset. In Dataset Split Robustness part of Section 5.4,

---

**Algorithm 2** Shift-resistant Bilevel Training

---

**Input:** EPOCH_1 and EPOCH_2
**Output:** Learned Embedding Space
 1: **for** out_loop_epoch $\leftarrow$ 1 to EPOCH_1 **do**
 2:    Forward propagate
 3:    **for** inner_loop_epoch $\leftarrow$ 1 to EPOCH_2 **do**
 4:       Optimize the sample weighting by Equation (1)
 5:    **end for**
 6:    Back propagate with the weighted joint loss $\mathcal{L} = \mathcal{L}_{est} + \gamma \mathcal{L}_{rec}$ to joint train
 7: **end for**

---

**Algorithm 3** Flatness Aware Gradient Ascent

---

**Input:** Initialized embedding $E$, LR bounds $\eta_1, \eta_2$, cycle length $c$, number of iterations $n$
**Output:** Optimal embedding $\hat{E}$
 1: $\hat{E} \leftarrow E$
 2: **for** $i = 1, 2, ..., n$ **do**
 3:    $\eta \leftarrow \eta(i)$  $\{$ Calculate LR for the iteration, $\eta_1 < \eta(i) < \eta_2$ $\}$
 4:    $E \leftarrow E + \eta \frac{\partial \omega_\theta}{\partial E}$  $\{$ Stochastic gradient update $\}$
 5:    **if** mod $(i, c) = 0$ **then**
 6:       $n_{\text{models}} \leftarrow \frac{i}{c}$  $\{$ Number of models $\}$
 7:       $\hat{E} \leftarrow \frac{\hat{E} \cdot n_{\text{models}} + E}{n_{\text{models}} + 1}$  $\{$ Update average $\}$
 8:    **end if**
 9: **end for**

---

we use five distinct splits on the Wine Quality White dataset to evaluate the proposed framework to respond to varying degrees of distributional shifts.

## C.2   MORE DETAILS FOR THE BASELINES

We compare our framework with 8 wildly-used feature transformation methods: 1) **Random Generation (RDG)** randomly produces feature-operation-feature transformations to generate a new feature space; 2) **Essential Random Generation (ERG)** initially expands the feature space by applying operations to each feature, and then chooses the essential features as the new feature space; 3) **Latent Dirichlet Allocation (LDA)** (Blei et al., 2003) refines the feature space to get the factorized hidden state via matrix factorization; 4) **AutoFeat Automated Transformation (AFAT)** (Horn et al., 2020) is an enhanced version of ERG, which repeatedly generates new features and employs multi-step feature selection to identify informative ones; 5) **Neural Feature Search (NFS)** (Chen et al., 2019) generates the transformation sequence for each feature and the entire process is optimized by RL; 6) **Traversal Transformation Graph (TTG)** (Khurana et al., 2018) formulates the transformation process as a graph and subsequently employs an RL-based search method to find the optimal feature set; 7) **Group-wise Reinforcement Feature Generation (GRFG)** (Wang et al., 2022) employs three collaborative reinforced agents to perform feature generation for feature space refinement. 8) **reinforceMent-enhanced autOregressive feAture Transformation (MOTA)** (Wang et al., 2024) formulates the discrete feature transformation problem as a continuous optimization task, while still assuming the I.I.D. condition.

## C.3   EVALUATION METRICS

To control the variance of the downstream model impact on evaluation, we apply Random Forests (RF) for both classification and regression tasks. For classification, we use the F1-score, Precision, and Recall as the evaluation metrics (Goutte & Gaussier, 2005). In regression, we assess performance using the following metrics (Hill, 2012; Hodson, 2022) 1 - Relative Absolute Error (1-RAE), 1 - Mean Absolute Error (1-MAE), and 1 - Mean Squared Error (1-MSE). For all these metrics, a higher value indicates a more effective feature transformation space.

Table 3: Time cost comparisons with baselines.

| Dataset | RL Data Collection (min) | RDG(s) | LDA(s) | ERG(s) | AFAT(s) | NFS(s) | TTG(s) | GRFG(s) | MOTA(min) | SAFT(min) |
|---|---|---|---|---|---|---|---|---|---|---|
| SpectF | 33.7 | 3.0 | 0.3 | 9.5 | 3.8 | 8.3 | 9.1 | 282.1 | 156.0 | 142.1 |
| Wine Quality White | 114.7 | 30.8 | 0.5 | 29.4 | 4.2 | 24.7 | 26.4 | 373.2 | 195.8 | 39.9 |
| Wine Quality Red | 36.5 | 7.4 | 0.6 | 13.9 | 3.6 | 9.2 | 10.3 | 182.2 | 107.5 | 43.1 |
| openml_616 | 156.6 | 27.5 | 0.2 | 55.9 | 2.6 | 18.1 | 24.6 | 936.5 | 430.1 | 160.7 |
| openml_618 | 239.6 | 67.7 | 0.2 | 82.4 | 3.0 | 30.9 | 38.7 | 821.6 | 373.8 | 95.9 |
| openml_637 | 171.3 | 28.1 | 0.2 | 55.8 | 2.8 | 19.3 | 25.1 | 731.7 | 300.9 | 147.9 |

Table 4: Space complexity comparisons with baselines.

| Dataset | RL Data Collection (MB) | RDG(MB) | LDA(MB) | ERG(MB) | AFAT(MB) | NFS(MB) | TTG(MB) | GRFG(MB) | MOTA(MB) | SAFT(MB) |
|---|---|---|---|---|---|---|---|---|---|---|
| SpectF | 0.19 | 0.13 | 0.07 | 0.14 | 0.08 | 0.14 | 0.13 | 1.04 | 0.14 | 0.16 |
| Wine Quality White | 0.21 | 0.15 | 0.08 | 0.16 | 0.09 | 0.15 | 0.14 | 18.70 | 0.13 | 0.48 |
| Wine Quality Red | 0.19 | 0.14 | 0.07 | 0.15 | 0.08 | 0.15 | 0.15 | 6.14 | 0.13 | 0.19 |
| openml_616 | 0.20 | 0.14 | 0.07 | 0.16 | 0.08 | 0.16 | 0.14 | 1.94 | 0.14 | 0.17 |
| openml_618 | 0.23 | 0.15 | 0.09 | 0.16 | 0.11 | 0.17 | 0.14 | 3.84 | 0.14 | 0.23 |
| openml_637 | 0.23 | 0.16 | 0.10 | 0.17 | 0.11 | 0.17 | 0.15 | 1.94 | 0.14 | 0.17 |

## C.4 HYPERPARAMETERS SETTING

1) RL collector: We use the reinforcement data collector to collect explored feature set-feature utility score pairs. The collector explores 512 episodes, and each episode includes 10 steps. 2) Graph Construction: The threshold for creating an edge between two features is the 95th percentile of all similarity values. 3) Our framework: we map the attribute of each node to a 64-dimensional embedding, and use a 2-layer GNN network and a 2-layer projection head to integrate such information. The decoder is a 1-layer LSTM network, which reconstructs a feature cross sequence. The evaluator is a 2-layer feed-forward network, in which the dimension of each layer is 200. During optimization, During optimization, we assign $\alpha$ (i.e., 10) as the weight for estimation loss and $\beta$ (i.e., 0.1) as the weight for reconstruction loss, the batch size is 256, the epochs are 500, and the learning rate range is 0.001-0.0005.

## D COMPARISON OF TIME AND SPACE COMPLEXITIES WITH BASELINE MODELS.

**Time complexity.** We analyzed the computational complexity of SAFT compared to baseline methods across six datasets. Table 3 shows that while RDG, LDA, ERG, AFAT, NSF, and TTG require less time, their accuracy is lower due to their inability to handle the shift problem. In contrast, our method offers lower costs and better performance compared to GRFG and MOTA. Additionally: 1) Feature transformation is neither a critical step in data preparation/processing nor a major time factor for most tasks; 2) The time cost of our method is acceptable, especially when compared to manual feature engineering, which can take days or months. Our approach significantly saves time; 3) Although our method may take more time than some baselines, it achieves superior feature engineering performance; 4) With our encoder-optimization-generation (EOG) design, we can further reduce training time by pre-training a foundational model and then fine-tuning it to other domains.

We also report the computational overhead of RL data collection, as shown in Table 3. While SAFT incurs additional time costs during data collection, this process is done asynchronously and offline, with an acceptable time cost. The main reason is that the RL-based collector requires more time to gather high-quality data, and the sequence formulas across the feature space increase the learning time of the sequence model.

**Space complexity.** To analyze the space complexity of SAFT, we illustrate the storage size required for different datasets. Table 4 shows that SAFT occupies minimal space and remains relatively stable. This is primarily due to the EOG framework, which embeds knowledge from variable-length discrete sequences into fixed-length embedding vectors. This embedding process keeps the parameter size stable, preventing it from increasing with data growth. Therefore, the experiments demonstrate that SAFT exhibits good scalability across datasets of different sizes.

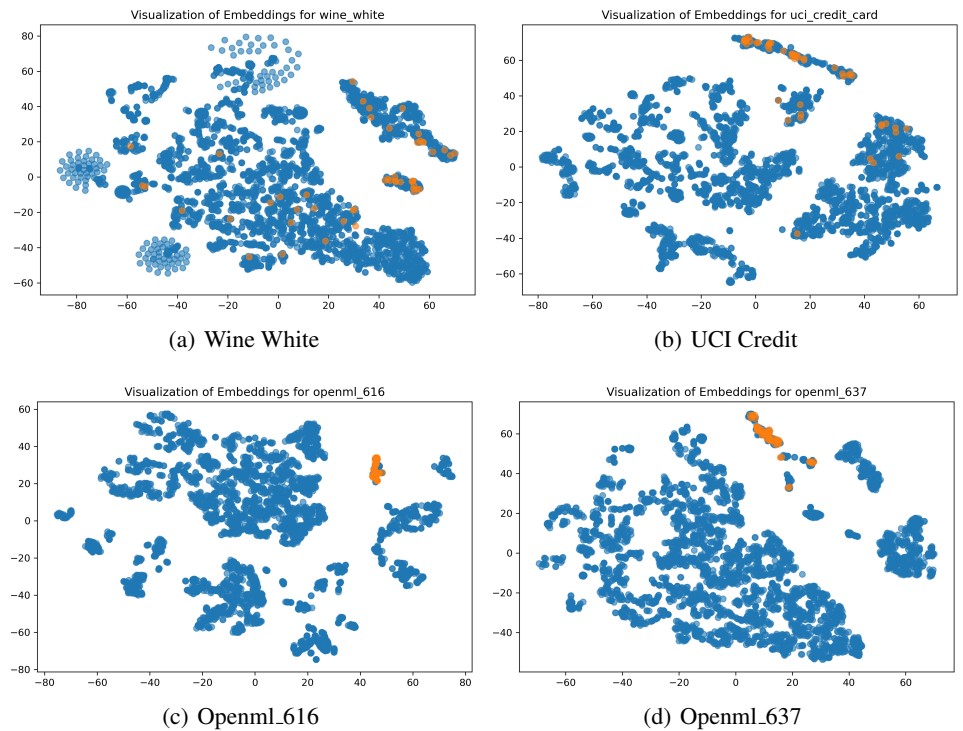

Figure 10: Visualization of the convergence embedding space. Each point represents a feature transformation operation sequence. The orange points represent the top 50 embedding points based on their downstream performance.

# E    VISUALIZATION FOR THE LEARNED EMBEDDING SPACE.

In this section, we visualized the embedding space to validate the effectiveness of the intermediate encoding. First, we collected the latent embeddings generated from the transformation records. Then, we used t-SNE to project these embeddings into a two-dimensional space for visualization. We selected four datasets for visualization as shown in Figure 10, where each point represents a feature transformation operation sequence. These points exhibit different distribution patterns, likely due to variations in the lengths of the corresponding transformation sequences, causing them to spread out in the embedding space. However, despite their differing positions, we observed that the points tend to cluster into several distinct groups, especially the points representing top-performing sequences (orange points), which tend to cluster closely together. This suggests that data points with strong downstream task performance are likely concentrated in certain regions, allowing gradient-based search algorithms to effectively locate optimal points within these areas when we use these points as initial search seeds. This case study highlights the role of continuous space in extracting feature knowledge through reconstruction and loss estimation, enabling the search for optimal embeddings to reconstruct the best feature transformation sequences.

