# OpenReview forum: "Distribution Shift Aware Neural Feature Transformation"
_ICLR.cc/2025/Conference — Submitted to ICLR 2025_

### Official Review · Reviewer_wzjW · 2024-10-29

**Soundness:** 3
**Presentation:** 3
**Contribution:** 2
**Rating:** 5
**Confidence:** 3

**Summary:**

This paper proposes a feature transformation technique to improve the AI capability from Shift-aware Representation-Generation perspective. And three mechanisms are proposed to address distribution shift by shift-resistant feature set representation, flatness-aware generation, and integration of normalization. Extensive experiments have been conducted on classification and regression tasks.

**Strengths:**

1. This paper is well presented and easy to understand.
2. This paper addresses an interesting topic, feature transformation is crucial to improve the performance of the model.

**Weaknesses:**

1. The contribution point 1 is not clear enough. The embedded-optimization-reconstruction framework was proposed in baseline method [1].  The distinction with baseline methods needs to be made clear in the motivation section.
2. The experimental results are questionable. In Table 1, some experimental results are not consistent with the results reported in [1], and are significantly lower than the reported results. For example, Higgs Boston, please give a reasonable explanation.  In addition, compared with the baseline method, there are still some results on other datasets not shown.
3. Ablation experiments. Three mechanisms proposed in the method, it is suggested to provide ablation experiments of these three mechanisms in the ablation experiments section.


[1] Reinforcement-Enhanced Autoregressive Feature Transformation: Gradient-steered Search in Continuous Space for Postfix Expressions. (NeurIPS 2023)

**Questions:**

The main questions are in weaknesses 1 to 3.

---

> ### Author Response · Authors · 2024-11-13
>
> We appreciate your acknowledgment that the paper is well-presented,  and the topic is crucial.
>
>
> Regarding your questions:
>
> ### W1:
>
> Our main contribution is addressing the distribution shift problem within the generative feature transformation framework. We re-summarize our contributions as follows:
> 1. We improved the embedded-optimization-reconstruction feature transformation framework to make it dataset distribution-aware rather than limited to token-level expressions.
> 2. We introduced a shift-resistant feature representation, flatness-aware generation, and incorporated normalization techniques to address distribution shift.
> 3. We conduct extensive experiments to demonstrate the efficiency, resilience, and traceability of our framework.
>
>
> ### W2:
>
> 1. Our reported baseline results appear lower due to different dataset splitting methods. [1] and other baseline papers use random sampling, ensuring the train and test sets follow the same distribution. In our experiments, we split the data to create a distinct distribution shift detected by Kolmogorov-Smirnov test between train and test sets.
> 2. We did not report results on other datasets because these datasets are stable and it's hard to observe a significant distribution shift when splitting the data. We will clarify this in the dataset section.
>
> ### W3:
>
> We propose addressing the OOD problem in feature transformation within a generative framework by introducing shift-resistant feature set representation, flatness-aware generation, and normalization integration. Ablation experiments in Section 5.2 demonstrate the effectiveness of these three components.
>
> We hope that our response can address your concerns.

---

> > ### Author Response · Authors · 2024-11-21
> >
> > Dear Reviewer,
> >
> > As the discussion period is swiftly approaching its endpoint. We would appreciate knowing if our responses have addressed your concerns. If we have sufficiently alleviated your concerns, we hope you'll kindly consider raising your score. Thank you for your time and consideration!

---

> > > ### Comment · Reviewer_wzjW · 2024-11-25
> > > **Distribution Shift Aware Neural Feature Transformation**
> > >
> > > Thanks for the rebuttal answer my questions.
> > >
> > > Since your main baseline method is [1], but using a different experimental setting than [1] is incomprehensible.  I remain skeptical of the experimental results.
> > >
> > > So I would keep my score.

---

> ### Author Response · Authors · 2024-11-25
>
> Thanks for acknowledging that our rebuttal has answered and addressed your questions.
>
> Regarding the final question about “why different experimental setting (total dataset number and ways of training/testing preparation) between our paper and [1]”:
>
> Because our AI task is distribution shift-resistant feature transformation, not classic feature transformation, our data environment of “using 16 datasets” and “creating training/testing shifts” rigorously strengthens the solidness and comprehensiveness,  instead of weakening experiments.
>
> 1. **why we SHOULD use 16 datasets instead of all 23 datasets in [1]:** When using our specialized sampling to create a distribution shift between training and testing data over the 23 datasets in [1], we found that the 16 of them show statistically-significant shifts rather than the other 7 datasets. We used the 16 drifted training/testing set pairs to ensure the fairness and solidness of our reported results. Instead, including the other 7 datasets without significant distribution shifts will reduce experimental solidness and fail to test shift resistance.
>
> 2. **why the baseline [1] in our paper’s setting SHOULD BE lower-performant than in [1]’setting under the same data source and same code:** We intentionally created distribution shifts between training and testing data. [1] reported classic feature transformation performance, but we reported shift-resistant performance under a drifted training/testing data environment.
>
> 3. **why our experiments are comprehensive:** Our experimental design (refer to paper and appendix) includes overall shift-resistant performance, impact of normalization, impact of flatness aware gradient search, impacts of reweighting, robustness check over diverse downstream predictors, robustness check over diverse strategies of shift creation, complexity analysis, visual and quality analysis of transformed features, visual and quality analysis of shift elimination.
>
> I hope the area chair can check our clarifications of the misunderstanding.

---

### Official Review · Reviewer_8d3t · 2024-10-30

**Soundness:** 2
**Presentation:** 2
**Contribution:** 2
**Rating:** 5
**Confidence:** 3

**Summary:**

This paper deals with the Neural Feature Transformation problem in the context of distribution shift, i.e., Distribution Shift Feature Transformation (DSFT) problem.

Specifically, it follows a representation-generation framework similar to (Wang et al., 2023), involving a representation step and a generation step. Moreover, three mechanisms are designed to deal with the DSFT problems: shift-resistant representation, flatness-awareness generation, and shift-aligned pre and post-processing.

During training, Shift-resistant Bilevel traning is proposed, and flatness-aware gradient ascent is incorporated.

Experiments are conducted on UCI and OpenML datasets with Kolmogorov-Smirnov to construct the distribution shifted data splits. Comparisons and ablations are conducted.

**Strengths:**

### Problem
- This paper addresses the Distribution Shifted Feature Transformation (DSFT) problem, which is relatively new and overall makes sense.
### Methodology & Ablation of Modules
- Specific techniques are designed to combat shifts in a representation-generation framework (Wang et al., 2023): bilevel training of samole weights and model parameters, flatness-aware gradien ascent.
The ablation in Figure 4 shows the effectiveness.
### Experiments
- Experiments are conducted on 16 benchmark datasets from UCI and OpenML. The proposed method outperforms SOTA in most settings.

**Weaknesses:**

### Organization and writing
- Overall, the main idea is easy to grasp. While the technical details, modules, and work flows are not quite easy to follow. For example, Algorithm 1 use full paragraph texts instead of algorithm-style to explain the framework.
- From line 627-632, the MOAT paper: Reinforcement-enhanced autoregressive feature transformation: Gradient-steered search in con-
tinuous space for postfix expression, appears twice. In line 294 argmin.
### Methodology
- While distribution shift problem is relatively novel, the main framework follows MOAT (Wang et al., 2023), see Figure 2 of MOAT. This is neither stated nor discussed in the paper. This is kind of misleading or overclaiming the contributions. A better practice is to explicitly clarify how the framework differes from MOAT, and the contributions to distribution shift.
- The RL pipeline for data preparation, $L_{rec}, L_{est}$ follow MOAT. Or any changes?
- New modules in Eq. (1) is straightforward with several existing work such as [R1]. The utilization for distribution shift and re-weighting is reasonable, but the overall technical contribution is not significant in this point.
- Flatness-aware gradient ascent is motivated from existing works (Izmailov et al., 2018; Garipov et al., 2018). According to Algorithm 3, it seems a simple technique.
### Experiments
- The SOTA baseline MOAT is experimented on 23 datasets, while this paper only on 16. Can you provide a justification why only 16 datasets are experimented or include more results for comprehensive comparison.


[R1] Learning to reweight examples for robust deep learning

**Questions:**

- What are the main differences of the proposed method and MOAT? And how these differences address the Distribution Shift?
- MOAT experimented on 23 datasets, while this paper only 16. Why the other benchmarks are not reported?

---

> ### Author Response · Authors · 2024-11-13
>
> We appreciate your acknowledgment that our research problem is new, our methodology is effective, and our experiments show a superior performance than baselines.
>
> Regarding your specific concerns:
>
> ### Organization and writing:
> **Q1:**
> We intended to highlight the insights of our approach at a high level, so we placed the more detailed algorithmic components in the appendix. We will reorganize this section in the camera-ready version.
>
> **Q2:**
> We removed the arxiv version.
>
> ### Methodology:
> **Q1:**
>  We reorganized our contributions and stated the difference between our model and MOAT:
> 1. MOAT didn't consider the OOD problem.
> 2. MOAT embeds the token-level expressions so it can't be aware of data distribution. Our method regards a feature set as a feature-feature interaction graph to capture the potential relationship between features and embeds raw data to be aware of the distribution of the dataset.
> 3. We introduced anti-shift mechanisms to address the OOD problem.
>
> Our contribution is summarized as:
> 1. We improved the embedded-optimization-reconstruction feature transformation framework to make it dataset distribution-aware rather than limited to token-level expressions.
> 2.  We introduced a shift-resistant feature representation, flatness-aware generation, and incorporated normalization techniques to address distribution shift.
> 3.  We conduct extensive experiments to demonstrate the efficiency, resilience, and traceability of our framework.
>
> **Q2:**
> Data preparation is not a crucial step in our framework. So we follow the advanced paper GRFG [1] to sample high-quality and diverse training data.
>
> [1] Wang, Dongjie, et al. "Group-wise reinforcement feature generation for optimal and explainable representation space reconstruction." Proceedings of the 28th ACM SIGKDD Conference on Knowledge Discovery and Data Mining. 2022.
>
> **Q3&Q4:**
> We address an essential AI task: The out-of-distribution problem in generative feature transformation, which is at the heart of ICLR. We propose a coherent approach to solve this problem. This is the first research task to solve the OOD of feature transformation. Several developed techniques in our method are generalizable:
> - shift-resistant feature set representation: it can rectify and adjust the distortion and ensure reliable and transferable feature knowledge in the testing set.
> - flatness-aware search: it's simple but effective to help identify the reliable in-distribution embedding.
> These computing thinking and insights are generic for feature engineering.
>
> ### Experiments:
> **Q1:**
> We did not report results on other datasets because these datasets are stable and it's hard to observe a significant distribution shift when splitting the data. We will clarify this in the dataset section.
>
>
> We hope that our response can address your concerns.

---

> > ### Author Response · Authors · 2024-11-21
> >
> > Dear Reviewer,
> >
> > As the discussion period is swiftly approaching its endpoint. We would appreciate knowing if our responses have addressed your concerns. If we have sufficiently alleviated your concerns, we hope you'll kindly consider raising your score. Thank you for your time and consideration!

---

> > > ### Comment · Reviewer_8d3t · 2024-11-25
> > > **Thanks for the rebuttal**
> > >
> > > Thanks for the rebuttal and address some of my questions.
> > > After carefully read the rebuttal and consider other comments, my concern regarding novelty and experiments remain.
> > > So I would keep my score.

---

> ### Author Response · Authors · 2024-11-25
>
> Thanks for acknowledging that our rebuttal has answered and addressed most of your questions, except “experimental data environment and novelty”.
>
> ### Regarding novelty:
>
> **Task novelty:** the tasks of  [1] and our paper are different: addressing feature transformation versus distribution-shift in feature transformation.
>
> **Technical novelty:** shift robustness via three mechanisms: feature transformation experience reweighing and flatness-aware search under generative learning, as well as pre-normalization and post-denormalization
>
> ### Regarding **why using 16 datasets instead of 23 datasets in [1]:**
>
> Because our AI task is distribution shift robustness in feature transformation, not classic feature transformation, our data environment of “using 16 datasets” and “creating training/testing shifts” rigorously strengthens the solidness and comprehensiveness,  instead of weakening experiments. This SHOULD NOT be a reason for rejection.
>
> 1. **why we SHOULD use 16 datasets instead of all 23 datasets in [1]:** When using our specialized sampling to create a distribution shift between training and testing data over the 23 datasets in [1], we found that the 16 of them show statistically-significant shifts rather than the other 7 datasets. We used the 16 drifted training/testing set pairs to ensure the fairness and solidness of our reported results. Instead, including the other 7 datasets without significant distribution shifts will reduce experimental solidness and fail to test shift resistance.
>
> 2. **why the baseline [1] in our paper’s setting SHOULD BE lower-performant than in [1]’setting under the same data source and same code:** We intentionally created distribution shifts between training and testing data. [1] reported classic feature transformation performance, but we reported shift-resistant performance under a drifted training/testing data environment.
> 3. **why our experiments are comprehensive:** Our experimental design (refer to paper and appendix) includes overall shift-resistant performance, impact of normalization, impact of flatness aware gradient search, impacts of reweighting, robustness check over diverse downstream predictors, robustness check over diverse strategies of shift creation, complexity analysis, visual and quality analysis of transformed features, visual and quality analysis of shift elimination.
>
> I hope the area chair can check our clarifications of the misunderstanding.

---

### Official Review · Reviewer_26Qx · 2024-11-01

**Soundness:** 2
**Presentation:** 3
**Contribution:** 2
**Rating:** 3
**Confidence:** 3

**Summary:**

Existing techniques for addressing the distribution shift cannot be directly applied to discrete search problems and present two primary challenges: (1) How can we reformulate and solve feature transformation as a learning problem? What mechanisms can integrate shift awareness into such a learning paradigm? To tackle these challenges, the authors leverage a unique Shift-aware Representation-Generation Perspective. To formulate a learning scheme, they construct a representation-generation framework: (1) representation step: encoding transformed feature sets into embedding vectors (2) generation step: pinpointing the best embedding and decoding as a transformed feature set. To mitigate the issue of distribution shift, they propose three mechanisms: (1) shift-resistant representation, where embedding dimension decorrelation and sample reweighting are integrated to extract the true representation that contains invariant information under distribution shift; (2) flatness-aware generation, where several suboptimal embeddings along the optimization trajectory are averaged to obtain a robust optimal embedding, providing effective for diverse distribution, and (3) shift-aligned pre and post-processing, where normalizing align and recover distribution gaps between training and testing data.

**Strengths:**

This paper addresses an interesting question: How do we transform features when there is a distribution shift?

**Weaknesses:**

- The formulation of the discrete search problem into a deep learning problem is similar to [Reinforcement-enhanced autoregressive feature transformation: Gradient-steered search in continuous space for postfix expressions].
- The novelty of the methodology is limited. Is the representation step specially designed for encoding feature transformation problems?
- Although considering the feature sets as a feature-feature interaction attributed graph is novel, the motivation behind it is not well explained. Why can the feature sets be considered a graph? Is it reasonable?
- Limited empirical validation. The authors only used 16 datasets for evaluation. However, as the baseline outlined in the paper [Reinforcement-enhanced autoregressive feature transformation: Gradient-steered search in continuous space for postfix expressions], there are 23 datasets that have been evaluated.

**Questions:**

See Weakness.

---

> ### Author Response · Authors · 2024-11-13
>
> We appreciate your acknowledgment that our research task is interesting and important.
>
> Regarding your specific concerns:
>
> ### W1:
>
> We improved the embedded-optimization-reconstruction feature transformation framework to make it encode raw data and aware of data distribution, rather than limited to token-level expressions. Then we developed anti-shift methods to address the OOD problem in the feature transformation task. We will reorganize these contributions and avoid misunderstanding.
>
> ### W2:
>
> 1. Our core task is to address the distribution shift problem in feature transformation.
> 2. We model the observed transformed feature set as a feature-feature interaction graph to capture underlying relationships between features and data distribution. We use inductive graph embedding algorithms to handle the dynamic changes in the number of nodes representing different observed transformed feature sets.
> 3. We use sample reweighting to debias shift.
>
>
> ### W3:
>
>
> Our main research task is to address the distribution shift problem in feature transformation. We regard the feature set as a graph for the following reasons:
>
> 1. To directly embed raw data and capture data distribution;
> 2. By representing the dataset as a feature-feature interaction graph, we can capture underlying relationships between features. Existing studies show that treating features or feature domains as nodes can effectively capture relationships between features.[1][2]
>
> [1] He, Xiaofei, Deng Cai, and Partha Niyogi. "Laplacian score for feature selection." Advances in neural information processing systems 18 (2005).
>
> [2] Li, Zekun, et al. "Fi-gnn: Modeling feature interactions via graph neural networks for ctr prediction." Proceedings of the 28th ACM international conference on information and knowledge management. 2019.
>
> ### W4:
>
> We did not report results on other datasets because these datasets are stable and it's hard to observe a significant distribution shift when splitting the data. We will clarify this in the dataset section.
>
> We hope that our response can address your concerns.

---

> > ### Author Response · Authors · 2024-11-21
> >
> > Dear Reviewer,
> >
> > As the discussion period is swiftly approaching its endpoint. We would appreciate knowing if our responses have addressed your concerns. If we have sufficiently alleviated your concerns, we hope you'll kindly consider raising your score. Thank you for your time and consideration!

---

> > > ### Comment · Reviewer_26Qx · 2024-11-25
> > >
> > > Dear Author
> > >
> > > Thanks for your clarification.
> > > It addresses some of my concerns, but I still think the experiments are not comprehensive enough.
> > > So, I would keep my score.
> > >
> > > Kind Regards

---

> > > > ### Author Response · Authors · 2024-11-25
> > > >
> > > > Thank you for acknowledging that my response has addressed most of your concerns.
> > > >
> > > > For the remained minor concern about experiment comprehensiveness:  **why using 16 datasets in our paper instead of 23 datasets in [1]:**
> > > >
> > > > Because our AI task is distribution shift robustness in feature transformation, not classic feature transformation, our data environment of “using 16 datasets” and “creating training/testing shifts” rigorously strengthens the solidness and comprehensiveness,  instead of weakening experiments. This SHOULD NOT be a reason for rejection.
> > > >
> > > > 1. **why we SHOULD use 16 datasets instead of all 23 datasets in [1]:** When using our specialized sampling to create a distribution shift between training and testing data over the 23 datasets in [1], we found that the 16 of them show statistically-significant shifts rather than the other 7 datasets. We used the 16 drifted training/testing set pairs to ensure the fairness and solidness of our reported results. Instead, including the other 7 datasets without significant distribution shifts will reduce experimental solidness and fail to test shift resistance.
> > > >
> > > > 2. **why the baseline [1] in our paper’s setting SHOULD BE lower-performant than in [1]’setting under the same data source and same code:** We intentionally created distribution shifts between training and testing data. [1] reported classic feature transformation performance, but we reported shift-resistant performance under a drifted training/testing data environment.
> > > >
> > > > 3. **why our experiments are comprehensive:** Our experimental design (refer to paper and appendix) includes overall shift-resistant performance, impact of normalization, impact of flatness aware gradient search, impacts of reweighting, robustness check over diverse downstream predictors, robustness check over diverse strategies of shift creation, complexity analysis, visual and quality analysis of transformed features, visual and quality analysis of shift elimination.
> > > >
> > > > I hope the area chair can check our clarifications of the misunderstanding.

---

### Official Review · Reviewer_Gywn · 2024-11-03

**Soundness:** 2
**Presentation:** 2
**Contribution:** 2
**Rating:** 5
**Confidence:** 3

**Summary:**

This paper studies feature transformation problem within the context of distributional shift in real-world scenarios. It introduces a Shift-aware Representation-Generation Perspective, which involves encoding transformed features into embedding vectors and decoding the optimal embedding. To address distributional shift, the paper proposes several techniques, including shift-resistant representation, flatness-aware generation, and shift-aligned pre- and post-processing. The effectiveness of these methods is evaluated through experiments on classification and regression tasks.

**Strengths:**

-	This paper addresses the important problem of feature transformation under distributional shift, a crucial challenge in deploying large foundation models in practical settings nowadays.
-	The paper introduces a comprehensive pipeline for feature transformation, including data collection, feature graph embedding, transformation, and pre-/post-processing stages.
-	Extensive experimental validation is provided, covering classification and regression on multiple datasets, ablation studies, and robustness analyses

**Weaknesses:**

-	The contribution of this paper appears somewhat ad hoc, as it incorporates a variety of techniques such as RL-based data collection, shift-resistant feature graph embedding, flatness-aware transformation, and shift-aligned pre-/post-processing without clearly explaining the motivation behind each component.
-	Several of the techniques used, particularly flatness-aware methods for addressing distributional shift, have been widely studied. The integration of these methods offers only incremental technical contributions.
-	The experiments are primarily conducted on small-scale datasets, raising concerns about the scalability and generalizability of the proposed approach to large-scale pretraining datasets nowadays.
-	The selection of baseline models for comparison appears outdated, as most were published prior to 2020, potentially limiting the relevance of the comparisons.
-	The paper's organization and clarity could be improved; for instance, the introduction contains redundant and hard-to-follow contexts that hinder readability.
-	The mathematical presentation is unsatisfactory; for example, providing a formal mathematical formulation for DSFT would enhance clarity over the current textual description.
-	Some notations are introduced before being defined, such as $f_1$, $f_2$.

**Questions:**

please refer to the weakness part.

---

> ### Author Response · Authors · 2024-11-13
>
> We appreciate your acknowledgment that our research problem is crucial, the proposed method is comprehensive, and the experiments are extensive.
>
> Regarding the weaknesses:
>
> ### W1:
>
> In each section, we emphasized the motivation behind each technique and we will further highlight this in the revised version.
> 1. Generative feature transformation framework: Prevents the exponential growth of possible feature combinations.
> 2. RL data collection: Leveraging RL’s exploration-exploitation mechanism automates the collection of high-quality, diverse training data to build a better continuous space.
> 3. Anti-shift mechanisms: In the generative feature transformation framework, training-testing shifts can distort the embedding space. Anti-shift mechanisms adjust for these distortions, ensuring reliable and transferable feature knowledge in the test set.
>
> ### W2:
>
> While some techniques we use, like flatness-aware methods, have been studied before, our integration of these techniques is a purposeful adaptation to tackle complex distributional shift challenges in feature transformation. Our experiments show that this approach effectively addresses OOD issues in feature transformation, which prior research has not considered. We believe this contribution is both innovative and practically impactful.
>
> ### W3:
>
> 1. The datasets are public and widely adopted for validating feature transformation approaches, as previous studies have extensively used them to demonstrate effectiveness.
> 2. Scalability is very important. Small-scale experiments are a valid first step for concept validation. Appendix D shows that our model performs well in terms of time and space complexity compared to advanced baselines like MOAT and GRFG, indicating its potential for scalability to larger datasets.
>
> ### W4:
>
> We used the most classic and commonly adopted baselines in feature transformation and also compared recent advanced models like GRFG and MOAT.
>
> ### W5:
>
> Our original purpose was to demonstrate our workload and respect to ICLR reviewers. This paper is intensive, but we put effort into writing and the writing includes our computing thinking. Our logic pipeline is:
> 1. background
> 2. limitations
> 3. highlight research gap
> 4. summarize our insights
> 5. summarize our proposed method
> 6. our contributions
>
> We will reorganize our introduction and remove some redundancy to make it more clear.
>
> ### W6:
>
> We provided a formal mathematical formulation for DSFT in the problem statement to enhance clarity:
> $$
>     \Gamma^{\ast}=\psi (\mathbf{E}^{\ast}) = \mathop{\arg\min}_{\mathbf{E}\in \varepsilon}\mathcal{A}(\mathcal{M}(\psi (\mathbf{E})(\mathbf{X}^{tr})), \mathbf{y}^{tr}),
> $$
> where $\varepsilon$ is the continuous embedding space by transforming $\mathbf{R}$. $\mathbf{E}$ is the embedding vector in the space of $\varepsilon$, and we define the optimal embedding vector is $\mathbf{E}^{\ast}$. $\psi$ is a reconstructing function that can rebuild the feature transformation operation sequence relying on the embedding vector $\mathbf{E}$ from $\varepsilon$. $\mathcal{A}$ indicates the effectiveness of the downstream task and the $\mathcal{M}$ represents the downstream model.
>
> Ultimately, we intend to employ $\Gamma^{\ast}$ to transform the original training feature set $\mathbf{X}^{tr}$ to the optimal training feature set $\mathbf{X}^{\ast}$. We expect the feature categories in $\mathbf{X}^{\ast}$ can also be the optimal feature categories on the test dataset $\mathcal{D}^{te}$ under distribution shifts.
>
> ### W7:
>
> We carefully checked these minor issues and fixed them.
>
> We hope that our response can address your concerns.

---

> > ### Author Response · Authors · 2024-11-21
> >
> > Dear Reviewer,
> >
> > As the discussion period is swiftly approaching its endpoint. We would appreciate knowing if our responses have addressed your concerns. If we have sufficiently alleviated your concerns, we hope you'll kindly consider raising your score. Thank you for your time and consideration!

---

> > > ### Comment · Reviewer_Gywn · 2024-11-28
> > >
> > > Thanks for the detailed response. I have also read the comments from other reviewers. However,  my concerns regarding the technical contribution, scalability, and generalizability of the proposed approach have not been adequately addressed in the author response. Therefore, I would like to keep my score.

---

### Meta-Review · Area_Chair_kYCM · 2024-12-17

**Metareview:**

The submission received the ratings of four reviewers, which recommended 5, 3, 5 and 5, averaging 4.50. Given the plenty of competitive submissions in ICLR, this stands at a score below the borderline. The reviewers' concerns remain on the ad-hoc novelty and the scalability of the experiments, and after the rebuttal, it seems that reviewers still maintain the concerns through their reply. Therefore, I tend to recommend rejection towards the current submission, and hope the challenging review helps the further improvement of the submission.

**Additional Comments On Reviewer Discussion:**

1. The ad-hoc technical novelty of the submission.

Not addressed.

2. The insufficient experiments, like 23 datasets in previous studies.

Not addressed.

---

### Decision · Program_Chairs · 2025-01-22

Reject